# Applicability of Pre-Plastic Deformation Method for Improving Mechanical Properties of Bulk Metallic Glasses

**DOI:** 10.3390/ma15217574

**Published:** 2022-10-28

**Authors:** Changshan Zhou, Hezhi Zhang, Xudong Yuan, Kaikai Song, Dan Liu

**Affiliations:** 1Shenzhen Research Institute of Shandong University, Shenzhen 518057, China; 2School of Mechanical, Electrical and Information Engineering, Shandong University, Weihai 264209, China; 3Department of Materials Science, University of Leoben, A-8700 Leoben, Austria; 4Weihai Wanfeng Magnesium Industry Science and Technology Development Co., Ltd., Weihai 264209, China

**Keywords:** bulk metallic glasses, pre-plastic deformation, Poisson’s ratio, shear bands, mechanical properties

## Abstract

Pre-plastic deformation (PPD) treatments on bulk metallic glasses (BMGs) have previously been shown to be helpful in producing multiple shear bands. In this work, the applicability of the PPD approach on BMGs with different Poisson’s ratios was validated based on experimental and simulation observations. It was found that for BMGs with high Poisson’s ratios (HBMGs, e.g., Zr_56_Co_28_Al_16_ and Zr_46_Cu_46_Al_8_), the PPD treatment can easily trigger a pair of large plastic deformation zones consisting of multiple shear bands. These PPD-treated HBMGs clearly display improved strength and compressive plasticity. On the other hand, the mechanical properties of BMGs with low Poisson’s ratios (LBMG, e.g., Fe_48_Cr_15_Mo_14_Y_2_C_15_B_6_) become worse due to a few shear bands and micro-cracks in extremely small plastic deformation zones. Additionally, for the PPD-treated HBMGs with similar high Poisson’s ratios, the Zr_56_Co_28_Al_16_ BMG exhibits much larger plasticity than the Zr_46_Cu_46_Al_8_ BMG. This phenomenon is mainly due to more defective icosahedral clusters in the Zr_56_Co_28_Al_16_ BMG, which can serve as nucleation sites for shear transformation zones (STZs) during subsequent deformation. The present study may provide a basis for understanding the plastic deformation mechanism of BMGs.

## 1. Introduction

In past decades, bulk metallic glasses (BMGs) have attracted widespread interest due to their outstanding mechanical properties [1,2,3,4,5,6], e.g., large elastic limit, high strength, high hardness, excellent fracture toughness, good corrosion resistance, good wear resistance, etc. However, due to the characteristics of long-range disorders and short-range orders in homogeneous macrostructures of BMGs, a highly localized plastic deformation via shear bands usually governs mechanical properties, resulting in catastrophic room-temperature brittleness [1,2,3,4,5,6], severely limiting future practical applications as engineering materials. In order to overcome catastrophic fracture and improve the plastic deformability of BMGs, various approaches have been suggested [7,8,9,10,11,12,13,14,15], which are classified into three basic strategies: The first is to create structural heterogeneity by tailoring chemical compositions, cooling rates, cryogenic thermal cycling, elastic loading, and so on [7,8]. The second is to introduce the second crystalline phase into a glassy matrix to prepare BMG composites by heating, manipulating the solidification process, or altering chemical compositions [9,10,11,12,13,14,15]. Among them, the BMG composites containing metastable B2 crystals with transformation-induced plasticity exhibit remarkable tensile plasticity and high strength with an obvious work-hardening effect [11,12,13,14,15]. The third is to change the deformation conditions or artificially produce macroscopic defects (e.g., notches, holes, shear bands) [16,17,18]. These methods above tremendously promote the development of BMGs and make BMGs research a hotspot in the material science community.

Furthermore, the pre-plastic deformation (PPD) method is frequently used to induce a certain amount of multiple shear bands in a glassy matrix and thus enhance mechanical properties, which involves the first and third strategies due to the micro-scale in length and nanoscale in width of induced shear bands together with structural heterogeneity brought by the rejuvenation via PPD treatments. Until now, improved mechanical properties in Zr-, ZrCu-, and Ti-based BMGs have been achieved through PPD methods (e.g., pre-compression, cold rolling, imprinting, etc.) [19,20]. Most early studies focused on the effect of the PPD on the solid-state amorphization process for binary and multicomponent glass-forming systems to illuminate glass formation or nanocrystalization behaviors [19,20]. Only a few investigations have attempted to improve the mechanical properties of amorphous ribbons or wires [21,22] due to the insufficient sample size. After entering the 21st century, Yokoyama et al. found that cold rolling can easily create multiple shear bands, which can effectively improve bending strength and plasticity [23]. At the same time, Jin et al. investigated the influence of pre-plastic deformation on the thermal stability of amorphous ribbons [24]. Afterwards, more observations on the influence of PPD treatments on thermal behaviors and mechanical properties were reported in Zr-based [25,26,27,28], revealing that PPD treatments improve both yield strength and compressive plasticity by introducing pre-existing shear bands. Furthermore, certain tensile plasticity is achieved in the cold-rolled Zr-based BMGs [25], while the fracture toughness is increased by 54% in the cold-rolled Zr_63.78_Cu_14.72_Ni_10_Al_10_Nb_1.5_ BMG. So far, most research objects are Zr-based alloy systems because these BMGs are more ductile [29,30,31]. However, it is still an open question whether the PPD treatment is applied to all the BMGs.

In addition, previous observations have demonstrated that the shape, number, and distribution of the pre-existing shear bands induced by PPD treatments, as well as the activated microstructural heterogeneity (e.g., free volume or shear transition zone (STZ)) in the plastic deformation zone, play a vital role in the enhancement of mechanical properties of BMGs [25,26,27,28]. Therefore, the effect of pre-existing shear bands on mechanical performance needs further investigation. In this study, the Zr_46_Cu_46_Al_8_, Zr_56_Co_28_Al_16_, and Fe_48_Cr_15_Mo_14_Y_2_C_15_B_6_ BMGs were selected to test the applicability of the PPD method on the mechanical properties of BMGs with high and low Poisson’s ratios (termed as HBMGs and LBMGs, respectively). Previous results have demonstrated that their corresponding Poisson’s ratios are ~0.366, ~0.367, and ~0. 309, respectively [29,32]. The mechanical properties and the shear banding behaviors after and before PPD treatments were investigated, and the potential deformation mechanism was also discussed.

## 2. Materials and Methods

Ingots with nominal compositions of the Zr_56_Co_28_Al_16_, Zr_46_Cu_46_Al_8_, and Fe_48_Cr_15_Mo_14_Y_2_C_15_B_6_ (at.%) were fabricated by arc melting pure elements (>99.9%) under Ti-gettered argon atmosphere, respectively. To ensure chemical homogeneity, the ingots were flipped and remelted at least three times during melting. The rods with a diameter of 2 mm were obtained using suction casting. The glassy structures were examined using X-ray diffraction (XRD, Rigaku D/max-rB, Cu *K_α_*, Wilmington, NC, USA), while the chemical composition distributions were observed by scanning electron microscopy (SEM, ZEISS Sigma500, Oberkochen, Germany) equipped with energy-dispersive X-ray spectrometry (EDS, OXFORD Ultim Extreme, High Wycombe, UK). The subsequent pre-deformation was conducted using an electronic tensile testing machine (SUNS UTM5015, Shenzhen, China) at an initial strain rate of 5 × 10^−4^ s^−1^. Figure 1a depicts a schematic diagram of the PPD processing on BMG rods. During the PPD treatment, a mechanical extensometer (NCS YYJ-4/10, Beijing, China) was used to measure the actual strain, and the final loading displacement was set as 0.1 ± 0.001 mm. For each experiment, two rods with the same length of ~4 mm were placed on the self-made tungsten steel gaskets and sent for the subsequent pre-deformation.

Uniaxial compression tests were conducted using an electronic tensile testing machine under a strain rate of 2.5 × 10^−4^ s^−1^. The dimension of the specimens for compression was Φ2 mm × 4 mm. The shear bands induced by PPD treatments were observed, and the surface morphologies after fracture were also investigated using SEM. Furthermore, Young’s modulus was measured using a nanoindentation device (Anton Paar CSM-NHT^2^, Ashland, VA, USA). During nanoindentation, a maximum force of 50 mN was achieved at a loading rate of 100 mN/min with a duration time of 10 s. Furthermore, the atomic structural features of the Zr_56_Co_28_Al_16_ BMG were observed by the ab initio molecular dynamics simulations using the Vienna ab initio simulation package (VASP) based on the density functional theory (DFT). The detailed simulation processing can be found in Reference [33].

## 3. Results

### 3.1. Pre-Existing Shear Bands in BMGs after PPD Treatments

Figure 1b exhibits the XRD patterns of the Zr_56_Co_28_Al_16_, Zr_46_Cu_46_Al_8_, and Fe_48_Cr_15_Mo_14_Y_2_C_15_B_6_ BMGs before and after PPD treatments. Only typical broad diffraction peaks are observed without any obvious crystalline diffractions.

As shown in Figure 1c–n, all elements are uniformly distributed in the glassy matrix for the Zr_56_Co_28_Al_16_, Zr_46_Cu_46_Al_8_, and Fe_48_Cr_15_Mo_14_Y_2_C_15_B_6_ BMGs. The corresponding average chemical compositions of the first two BMGs were measured as Zr_56.2_Co_28.0_Al_15.8_ and Zr_46.6_Cu_45.3_Al_8.1_, respectively. Since it is impossible to measure elements with small atomic numbers accurately by EDS, the average composition of Fe-based BMG was determined to be Fe_50.2_Cr_15.3_Mo_20.7_Y_3.3_C_15.3_ BMGs without element B. These observations confirm their homogeneous amorphous structure. Figure 2 shows the SEM images of the Zr_56_Co_28_Al_16_, Zr_46_Cu_46_Al_8_, and Fe_48_Cr_15_Mo_14_Y_2_C_15_B_6_ BMGs after PPD treatments. It can be found that a pair of plastic deformation regions (green dotted circles) appears on almost parallel sides of both Zr_56_Co_28_Al_16_ and Zr_46_Cu_46_Al_8_ BMGs (Figure 1a,f), which display a very large hemispherical shape. Obviously, the deformation morphology is governed by the radial shear bands, which is very similar to the plastic flow zone achieved during Vickers indentation for BMGs [34]. The enlarged SEM images of these plastic deformation regions (i.e., R1–R4) are shown in Figure 1b,c,e,f. It is clearly seen that the plastic deformation zone consists of two types of shear bands, i.e., relatively long semi-circular (red dotted lines) shear bands and dense wavy (red arrows) shear bands that appear between semi-circular ones. During deformation, the initial elastic and plastic deformations first concentrate on two tiny plastic zones at both the top and bottom of HBMG rods. When the plastic deformation occurs at both contact regions, the shear bands (yellow dash-dotted lines) initiate at both contact areas with the help of the normal stress upon loading. At the same time, the neighboring regions should bear significant horizontal pressure when the semi-circular contact regions gradually become flat. A resultant force does not strictly follow and starts to depart from the loading direction, while the deviation degree increases when the plastic deformation zones become larger. As a result, shear bands will not be straight but semi-circular. Meanwhile, these shear bands also interact with each other, leading to the appearance of dense wavy shear bands. On the contrary, only very small plastic deformation zones (green dotted circle) appear at the top and bottom of the Fe_48_Cr_15_Mo_14_Y_2_C_15_B_6_ LBMG (Figure 2g). Take the plastic deformation zone R5 for an example, and the corresponding enlarged view was shown in Figure 2h. The initial semi-circular shear bands (dotted red lines) appear and accumulate at the top and bottom of LBMG rods. The subsequent shear banding propagation does not occur after the LBMG is subjected to a loading displacement of ~0.1 mm. However, some micro-cracks appear along the induced shear bands (solid yellow arrows in Figure 2i). These observations indicate that it becomes more difficult for LBMGs to be deformed than HBMGs.

### 3.2. Mechanical Properties of BMGs in As-Cast States and after PPD Treatments

Figure 3 shows the mechanical properties of the Zr_46_Cu_46_Al_8_, Zr_56_Co_28_Al_16_, and Fe_48_Cr_15_Mo_14_Y_2_C_15_B_6_ BMGs in the as-cast state and after PPD treatments. Before PPD treatments, the mechanical properties are different even though the as-cast HBMGs with a similar high Poisson’s ratio. The yield strength, maximum compressive strength, and plastic strain of the as-cast Zr_56_Co_28_Al_16_ HBMG are measured to be 2115 ± 10 MPa, 2168 ± 20 MPa, and ~4.4%, while those of the as-cast Zr_46_Cu_46_Al_8_ HBMG are 1992 ± 10 MPa, 1995 ± 20 MPa, and ~0.2%, respectively (Figure 3a,b and Table 1). Moreover, the as-cast Fe_48_Cr_15_Mo_14_Y_2_C_15_B_6_ LBMG shows a representative brittle behavior, and no plasticity is observed (Figure 3c), whose fracture strength is 3239 ± 20 MPa. However, as shown in Figure 3a,b, the mechanical properties of both Zr_46_Cu_46_Al_8_ and Zr_56_Co_28_Al_16_ HBMGs after PPD treatments are enhanced. After PPD treatments, the yield strength and maximum compressive strength of the Zr_56_Co_28_Al_16_ HBMG are enhanced to be 2122 ± 15 MPa and 2420 ± 30 MPa, while those of the Zr_46_Cu_46_Al_8_ HBMG are improved to be 1992 ± 15 MPa and 2120 ± 30 MPa, respectively (Figure 3a,b and Table 1). It is worth noting that the enhancement of yield strength of both Zr_46_Cu_46_Al_8_ and Zr_56_Co_28_Al_16_ HBMGs is very limited, approximately 0.33% and 0.25%, respectively. However, the maximum compressive strength is distinctly improved, while the increased proportion is about 11.6% and 6.3% for the Zr_46_Cu_46_Al_8_ and Zr_56_Co_28_Al_16_ HBMGs, respectively. It has been elucidated that the enhanced strength should be attributed to the residual stress upon pre-deformation [35,36].

Moreover, the enhancement of plasticity is also remarkable for both HBMGs. The plastic strain of the Zr_56_Co_28_Al_16_ HBMG increases from ~4.4% to ~21.8%, while that of the Zr_46_Cu_46_Al_8_ HBMG rises from 0.2% to ~3.5%. Even though both BMGs show a similar Poisson’s ratio, the improved mechanical performance of the Zr_56_Co_28_Al_16_ HBMG is far more significant than that of the Zr_46_Cu_46_Al_8_ HBMG after PPD treatments. On the other hand, the mechanical properties of the Fe_48_Cr_15_Mo_14_Y_2_C_15_B_6_ LBMGs become worse after PPD treatments, whose fracture strength decreases to 2909 ± 30 MPa without detectable plasticity.

### 3.3. Fracture Morphologies

As shown in Figure 4a, only several shear bands are observed along the fracture plane for the as-cast Zr_46_Cu_46_Al_8_ HBMG, while multiple shear bands appear after PPD treatments (Figure 4b). The newly formed shear bands initiate from rod surfaces where the PPD treatment was conducted (region A in Figure 4b) and then propagate parallel to the main shear band. For the as-cast Zr_56_Co_28_Al_16_ HBMG, a lot of shear bands appear after failure without bearing PPD treatments (Figure 4c). After PPD treatment, more multiple shear bands form in the Zr_56_Co_28_Al_16_ HBMG than in the Zr_46_Cu_46_Al_8_ HBMG. More importantly, some shear bands cross with each other along ~43º, while other dense shear bands perpendicular to the loading direction exist between the crossed shear bands (Figure 4d), implying that the newly formed shear bands induced by PPD treatments play a vital role in the subsequent plastic deformation.

However, for the Fe_48_Cr_15_Mo_14_Y_2_C_15_B_6_ LBMGs, the samples are broken down into pieces and exhibit typical brittle features (Figure 5) in the as-cast state and after PPD treatments. These observations imply that the plastic deformation of the Fe_48_Cr_15_Mo_14_Y_2_C_15_B_6_ LBMGs cannot be affected by the pre-existing shear bands induced by PPD treatment and is still governed by crack propagation. The existence of micro-cracks induced by PPD treatments can provide crack-initiation sites for subsequent crack-linking and propagation.

## 4. Discussion

Compared to the Fe_48_Cr_15_Mo_14_Y_2_C_15_B_6_ LBMG, multiple shear bands govern the whole plastic deformation of both Zr_46_Cu_46_Al_8_ and Zr_56_Co_28_Al_16_ HBMGs. As we know, the formation of shear bands is caused by the percolation of a serial of the activated shear transformation zones (STZs) [1,2,3,37,38,39]. The STZ is one local cluster of atoms that experiences inelastic shear distortion by conquering the activation energy barrier between two different energy configurations [1,2,3,37,38,39]. During deformation, the nucleation of a shear band is believed to be controlled by the activation of local STZs at looser atomic structures. The activation of local STZs proceeds through two steps [1,2,3,37,38,39,40], i.e., a flow-induced dilatation that broadens homogeneously during the early stage and the localization of plastic strain into a narrow shear band at the expense of shear flow in the surrounding regions. With further deformation, the shear banding propagation is closely linked with the percolated connection and the cooperative shear of a large amount of local STZs. So far, many investigations have been conducted to evaluate STZ sizes or volumes in BMGs [40,41,42], among which a cooperative shear model is widely accepted to calculate them [43]. The prevailing view is that a large STZ volume facilitates the plasticity of BMGs [44,45,46]. Moreover, a high Poisson’s ratio or a low ratio of shear modulus (*G*) and bulk modulus (*B*) may be responsible for good plasticity of BMGs [47]. The critical Poisson’s ratio for the ductile–brittle transition was determined to be 0.31~0.32 [47]. Thus, the STZ volume, Poisson’s ratio, and plasticity are usually correlated in different BMGs.

Pan et al. found that the STZ volume increases with increasing Poisson’s ratio, leading to the improved plasticity of BMGs [48], while Qiao et al. believed that this rule is no longer applicable [49]. Herein, we collected the reported STZ volumes and Poisson’s ratios of BMGs, which are shown in Figure 6a. It is obvious that the STZ volumes roughly increase with increasing Poisson’s ratios for the reported BMGs (detailed data listed in the Appendix A). After carefully examining previous observations [42,50,51,52], the STZ volumes strongly depend on the loading conditions (e.g., strain rates, applied pressure, tip radius, etc.), experimental technologies, calculation methods, and so on. For instance, in Zr-based BMGs, simulations and experimental observations based on the statistical analysis of the first pop-in data give STZ volumes of approximately 0.2~0.4 nm^3^ [53], while the STZ volumes calculated based on the rate-dependent hardness data are estimated to be 1~16 nm^3^ [46,54,55,56,57] (Figure 6a). As shown in Figure 6b, when the calculation method is fixed, the STZ volume exhibits a strong loading rate-dependence or tip radius-dependence during nanoindentation. Moreover, the Poisson’s ratios of some BMGs are mismeasured, leading to different Poisson’s ratios for a same glass-forming composition. As a result, the correlation between the STZ volume and Poisson’s ratio does not show a good change tendency. When the calculation methods or experimental processing are fixed, the plasticity indeed increases with increasing STZ volumes (Figure 6c). Furthermore, PPD treatments (e.g., cold rolling) on BMGs indeed show the increase in the released enthalpy prior to the glass transition temperature (*T_g_*) (Figure 6d), which reflects the increase in the free volume. Since the liquid-like regions containing high local free volume provide potential sites for the STZ operation, the increased free volume can cause the easy nucleation and activation of STZs [42,58]. In our case, compared with the Fe_48_Cr_15_Mo_14_Y_2_C_15_B_6_ LBMG, both Zr_46_Cu_46_Al_8_ and Zr_56_Co_28_Al_16_ LBMGs exhibit a higher Poisson’s ratio and then a potential larger STZ volume, leading to the easier initiation of shear bands during PPD treatments. Therefore, more STZs will be easily activated in the PPD-treated Zr_46_Cu_46_Al_8_ and Zr_56_Co_28_Al_16_ glassy matrix during compression, especially in the previous plastically deformed regions. As a result, more multiple shear bands appear, resulting in enhanced plasticity.

However, the Zr_56_Co_28_Al_16_ HBMG exhibits better mechanical properties than the as-cast Zr_46_Cu_46_Al_8_ in the as-cast state and after PPD treatments, respectively, even though both Poisson’s ratios and STZ volumes are at the same level size. It is noted that the STZs are not structural defects in BMGs, but an event defined in a local volume that is also strongly influenced by local atomic arrangements [39]. The shear banding process depends on the nucleation and activation of local STZs and the subsequent percolation and propagation of STZs, which should be strongly linked to the local atomic structures of BMGs [47,62]. Therefore, the difference between local atomic structures of both Zr_46_Cu_46_Al_8_ and Zr_56_Co_28_Al_16_ HBMGs should be crucial in determining the multiplication of shear bands and ultimately manipulating plastic deformation behaviors. Fang et al. simulated the atomic configuration of the as-cast Cu_46_Zr_46_Al_8_ HBMG by the ab initio molecular dynamics simulations and investigated its local atomic packings by virtue of the Voronoi tessellation method [63]. It was found that the distribution of polyhedra in the Zr-centered architectures is relatively dispersive and the predominant polyhedra are the <0,2,8,4>, <0,1,10,4>, <0,1,10,3>, <0,2,8,5>, and <0,1,10,2>, whose corresponding fractions are measured to be ~6.55%, ~6.35%, ~5.93%, ~5.40%, and ~3.73%, respectively [63]. Most polyhedra in the Zr-centered architectures belong to large Kasper polyhedra with coordination numbers. In the Cu-centered architectures, the polyhedra are mainly governed by the 15.5% ⟨0,2,8,1⟩, 9.0% ⟨0,2,8,2⟩, 7.5% ⟨0,0,12,0⟩, and 7.0% ⟨0,3,6,3⟩. Beside the perfect ⟨0,0,12,0⟩ polyhedron, the ⟨0,2,8,1⟩, ⟨0,2,8,2⟩, and ⟨0,3,6,3⟩ can be treated as defective icosahedra [63]. Therefore, the Cu atoms with the smallest atomic size are mainly surrounded by icosahedral type (defective or full) clusters. In the Al-centered architectures, the full icosahedron represented by ⟨0,0,12,0⟩ takes a remarkable percentage of ~30.0%, while the ⟨0,3,6,3⟩ icosahedron displays a fraction of ~14.3%, which distribute as a “backbone” structure [63].

In order to illustrate the local atomic packings of the as-cast Zr_56_Co_28_Al_16_ HBMG, the ab initio MD calculation was also performed, whose polyhedral distributions are shown in Figure 7. It can be seen from Figure 7a that the predominant polyhedra in the Zr-centered architectures are the <0,2,8,5>, <0,1,10,4>, <0,1,10,3>, <0,2,8,4>, <0,3,6,4>, <0,3,6,5>, <0,3,6,6>, and <0,1,10,2>, whose percentages were calculated to be ~8.85%, ~8.56%, 8.~56%, ~7.41%, ~6.50%, ~6.25%, ~3.82%, and ~3.78%, respectively. Being in line with the observations on Cu_46_Zr_46_Al_8_ HBMG, most polyhedra in the Zr-centered architectures are large Kasper polyhedra with coordination numbers (Figure 7b). In the Co-centered architectures, the governed polyhedra are found as follows: <0,3,6,1>, <0,3,6,0>, <0,2,8,0>, <0,2,8,1>, <0,4,4,3>, and <0,2,8,2>, whose factions are approximately determined to be ~20.5%, ~14.4%, ~13.48%, ~9.4%, ~6.9%, and ~6.7%, respectively. Hence, the Co atoms are mainly surrounded by defective icosahedral-type clusters. In the Al-centered architectures, the predominant polyhedra are dominated by ~18.8% <0,1,10,2>, ~17.1% <0,2,8,1>, ~11.3% <0,2,8,2>, and ~6.1% <0,3,6,2> together with ~6.4% <0,0,12,0> full icosahedron (Figure 7c). Compared to the Cu_46_Zr_46_Al_8_ BMG, the Co- and Al-centered architectures in the as-cast Zr_56_Co_28_Al_16_ BMG contains more defective icosahedral clusters, which should provide more nucleation sites for STZs during deformation. Therefore, even though PPD treatments generate similar shear bands in both Zr_46_Cu_46_Al_8_ and Zr_56_Co_28_Al_16_ HBMGs before the final compression test, more multiple shear bands should be induced in the PPD-treated Zr_56_Co_28_Al_16_ BMG, leading to larger plasticity of the Zr_56_Co_28_Al_16_ HBMG.

## 5. Conclusions

In this work, three types of BMGs with different Poisson’s ratios, i.e., Zr_46_Cu_46_Al_8_, Zr_56_Co_28_Al_16_, and Fe_48_Cr_15_Mo_14_Y_2_C_15_B_6_ BMGs, were chosen to verify the impact of the pre-plastic deformation on the generation of multiple shear bands in BMGs. According to the XRD and EDS observations, all the samples are still amorphous, and no nanocrystallization appears after the pre-plastic deformation via the current sideloading method. For the Zr_46_Cu_46_Al_8_ and Zr_56_Co_28_Al_16_ BMGs with high Poisson’s ratios, a pair of larger plastic deformation zones can be induced at loading positions after pre-plastic deformation. Many long semi-circular shear bands appear within both plastic deformation zones, while some dense wavy shear bands emerge between these long shear bands. The dominant reason is that the resultant force changes continuously when the pre-plastic deformation proceeds. For the Fe_48_Cr_15_Mo_14_Y_2_C_15_B_6_ BMG with a high Poisson’s ratio, only very small plastic deformation zones are caused after the pre-plastic deformation. Within these small zones, only a few semi-circular shear bands appear, accompanied by several micro-cracks.

Then the room-temperature compression tests were conducted on these pre-deformed BMG rods to measure their mechanical properties. It was observed that those BMGs with high Poisson’s ratios show higher strength and larger plasticity after pre-plastic deformation. However, the Fe_48_Cr_15_Mo_14_Y_2_C_15_B_6_ BMG with a high Poisson’s ratio after the pre-plastic deformation exhibits worse mechanical properties than the as-cast state, which should be attributed to the relatively small STZ volume because of a low Poisson’s ratio. Additionally, despite the Poisson’s ratios of the Zr_56_Co_28_Al_16_ and Zr_46_Cu_46_Al_8_ BMGs being similar, the improvement of mechanical properties of the Zr_56_Co_28_Al_16_ BMG is more evident than those of the Zr_46_Cu_46_Al_8_ BMG. Molecular dynamics simulation observations indicate that the Co- and Al-centered architectures in the as-cast Zr_56_Co_28_Al_16_ BMG contain more defective icosahedral clusters, which can provide more nucleation sites for STZs during deformation and lead to better mechanical properties.

## Figures and Tables

**Figure 1 materials-15-07574-f001:**
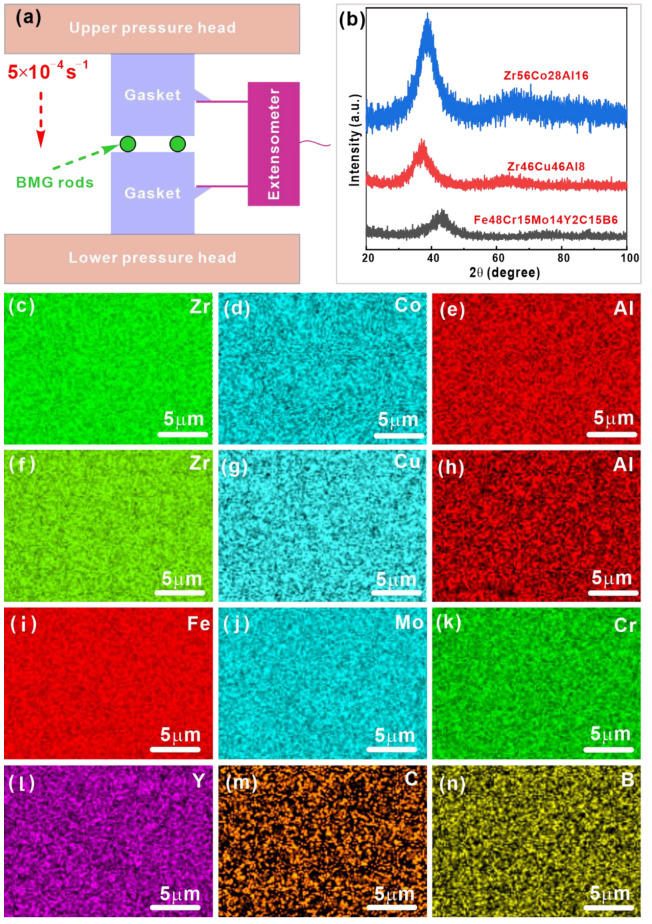
(**a**) Schematic diagram of the PPD processing on BMG rods, (**b**) XRD patterns of BMGs after plastic deformation, EDS mapping results of (**c**–**e**) Zr_56_Co_28_Al_16_, (**f**–**h**) Zr_46_Cu_46_Al_8_, and (**i**–**n**) Fe_48_Cr_15_Mo_14_Y_2_C_15_B_6_ BMGs.

**Figure 2 materials-15-07574-f002:**
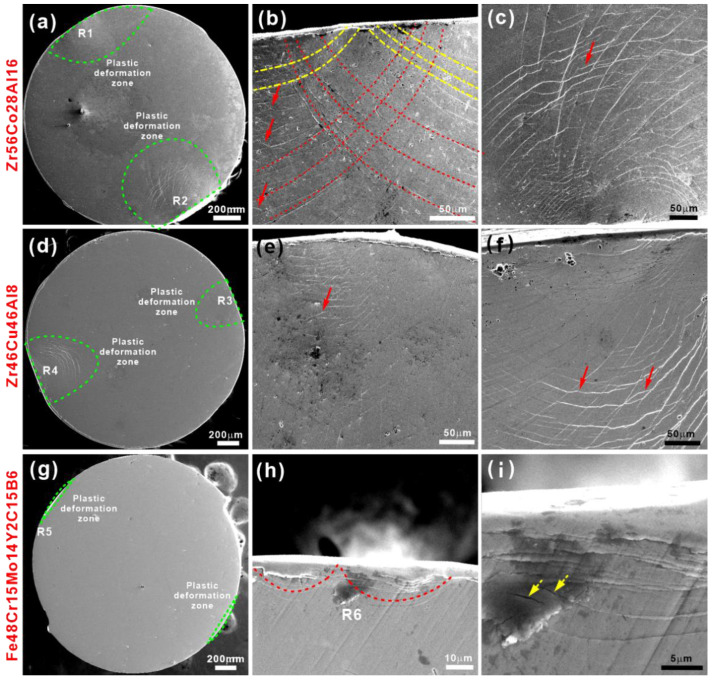
Overall morphologies of the (**a**) Zr_56_Co_28_Al_16_, (**d**) Zr_46_Cu_46_Al_8_, and (**g**) Fe_48_Cr_15_Mo_14_Y_2_C_15_B_6_ BMGs after pre-plastic deformation and the enlarged SEM images of the regions (**b**) R1, (**c**) R2, (**e**) R3, (**f**) R4, (**h**) R5, and (**i**) R6, respectively; The green dotted lines represent the plastic deformation zone induced by PPD treatments. The yellow dot–dash lines represent the semi-circular shear bands developed from the initial shear bands induced by PPD treatments. In contrast, the red dotted lines represent the induced semi-circular shear bands during the later stage of PPD treatments. The solid red arrows represent the dense wavy shear bands between semi-circular ones, and the dotted yellow arrows represent micro-cracks.

**Figure 3 materials-15-07574-f003:**
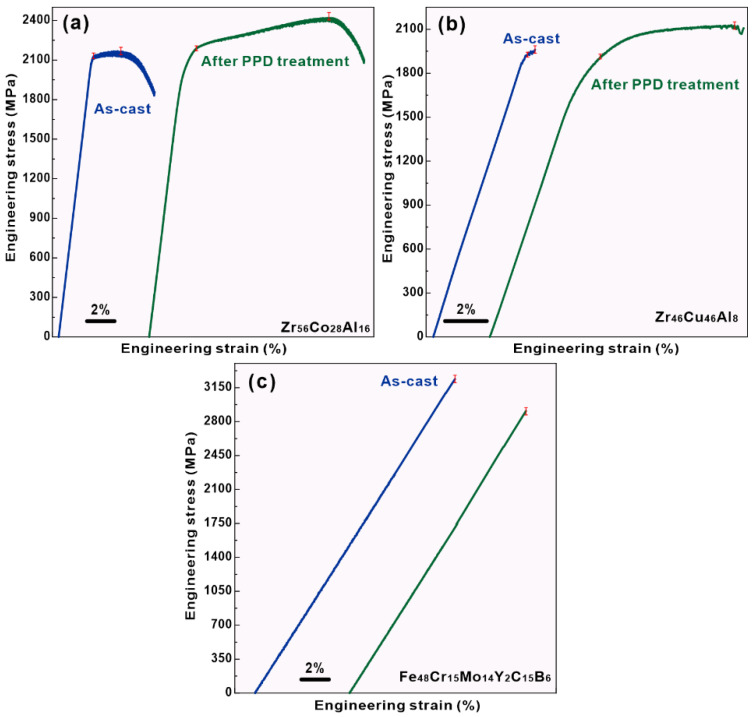
Engineering stress-strain curves under compression: (**a**) Zr_56_Co_28_Al_16_, (**b**) Zr_46_Cu_46_Al_8_, and (**c**) Fe_48_Cr_15_Mo_14_Y_2_C_15_B_6_ BMGs in the as-cast state and after PPD treatments.

**Figure 4 materials-15-07574-f004:**
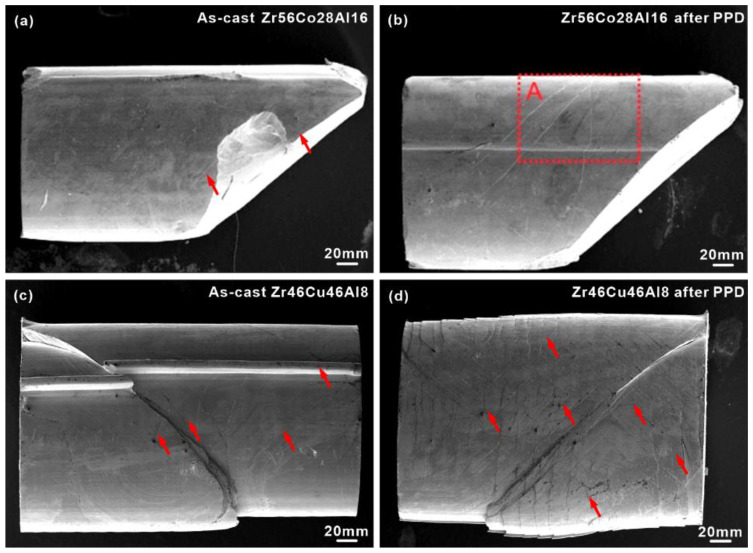
Fracture morphologies of the (**a**,**b**) Zr_46_Cu_46_Al_8_ and (**c**,**d**) Zr_56_Co_28_Al_16_ BMGs in the as-cast state and after PPD treatments; the solid arrows and dotted square A represent shear bands and the regions where shear bands exist.

**Figure 5 materials-15-07574-f005:**
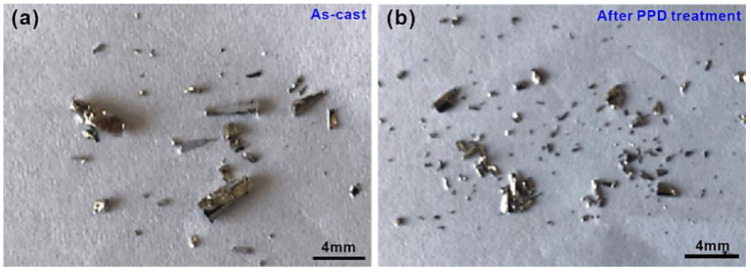
Fracture morphologies of the Fe_48_Cr_15_Mo_14_Y_2_C_15_B_6_ BMG (**a**) in the as-cast state and (**b**) after PPD treatments.

**Figure 6 materials-15-07574-f006:**
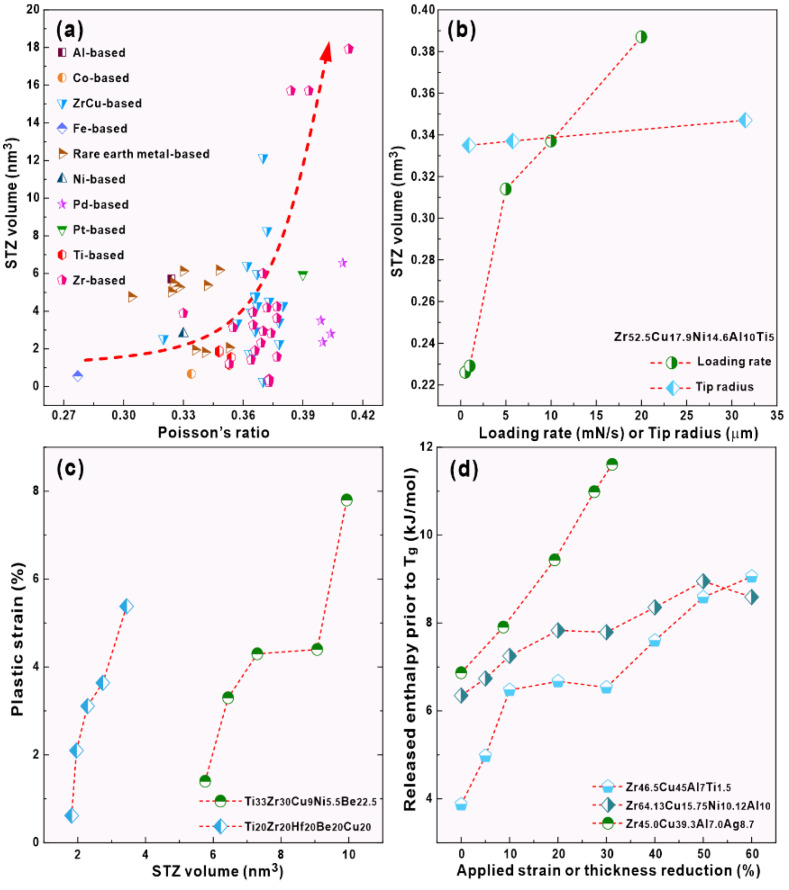
(**a**) Correlation between the STZ volumes and Poisson’s ratios in different systems BMGs (Red arrow indicates the increasing tendency), (**b**) the change in the STZ volumes with increasing loading rate or tip radius during nanoindentation [42,53], (**c**) correlation between the STZ volumes and plastic strains [54,59], and (**d**) the increased enthalpy before *T_g_* after cold rolling [60,61].

**Figure 7 materials-15-07574-f007:**
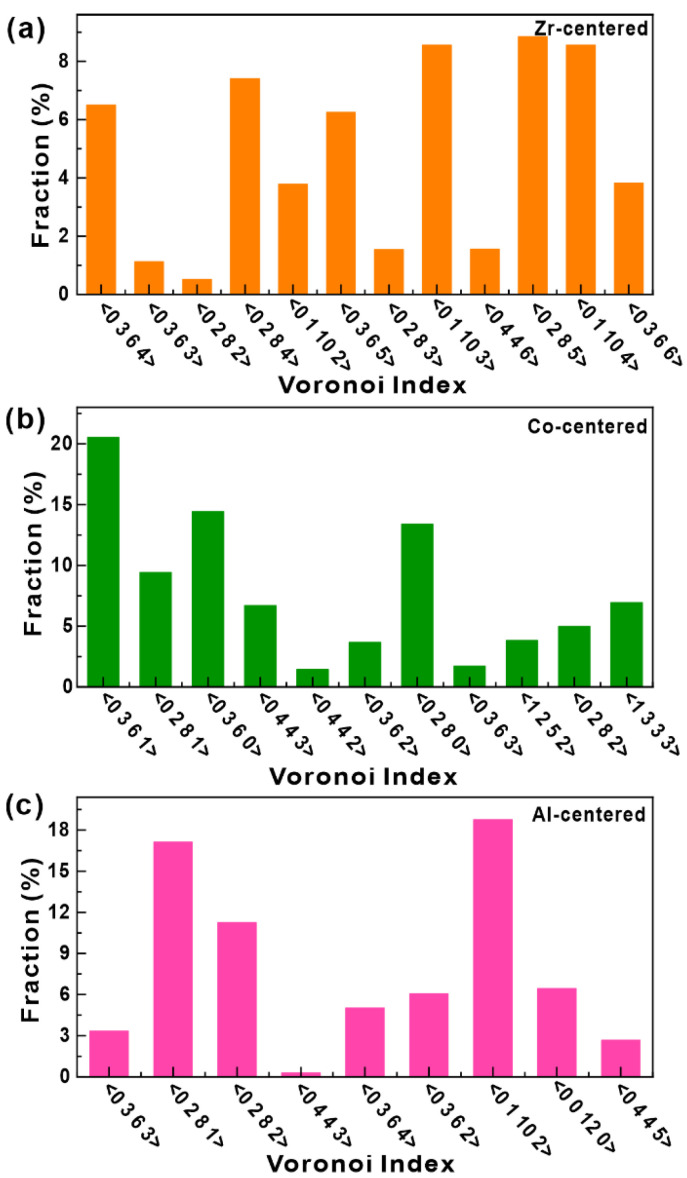
Distributions of Voronoi polyhedron centered by the (**a**) Zr, (**b**) Co, and (**c**) Al atoms, respectively.

**Table 1 materials-15-07574-t001:** Poisson’s ratio, yield strength, and plasticity of Zr_56_Co_28_Al_16_, Zr_46_Cu_46_Al_8_, and Fe_48_Cr_15_Mo_14_Y_2_C_15_B_6_ BMGs in the as-cast state and after PPD treatments.

Compositions	State	Poisson’s Ratio	Yield Strength (MPa)	Max. Compressive Strength (MPa)	Plasticity (%)
Zr_56_Co_28_Al_16_	cast	~0.367	2115 ± 10	2168 ± 20	~4.4
Zr_56_Co_28_Al_16_	PPD	-	2122 ± 15	2420 ± 30	~21.8
Zr_46_Cu_46_Al_8_	cast	~0.366	1990 ± 10	1995 ± 20	~0.2
Zr_46_Cu_46_Al_8_	PPD	-	1992 ± 15	2120 ± 30	~3.5
Fe_48_Cr_15_Mo_14_Y_2_C_15_B_6_	cast	~0.309	-	3239 ± 20	0
Fe_48_Cr_15_Mo_14_Y_2_C_15_B_6_	PPD	-	-	2909 ± 30	0

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
