# Peer review of "Applicability of Pre-Plastic Deformation Method for Improving Mechanical Properties of Bulk Metallic Glasses"

_materials, 2022, doi:10.3390/ma15217574_

Round 1

Reviewer 1 Report

This paper investigated the applicability of the PPD method in three different BMGs (Zr46Cu46Al8, Zr56Co28Al16, and Fe48Cr15Mo14Y2C15B6 BMGs) with different Poisson’s ratios and attempt to enhance the strength and compressive plasticity of PPD-treated HBMGs.

Results

1.      Figure 2, add more detail to what red and yellow lines represent as well as the red arrow in the Fig. caption.

2.      XRD analysis was mentioned in the methodology but results not presented. To confirm that, the fabricated BMGs before and after treatment are still amorphous, XRD spectra must be included. This is very critical!

3.      EDX, XRF or XPS analysis must be included to confirm the elemental compositions of each BMGs

Conclusion:

4.      Check line 5 “…. due to theesultant …..” kindly check the spelling.

Author Response

Dear Editors and Reviewers,

The authors sincerely thank the editors again for dealing with our manuscript. We also highly appreciate the reviewers’ constructive comments and valuable suggestions. Those comments are very valuable and helpful for improving our work. Now the XRD and EDX observations were added.

In the following, the comments were listed in italic blue font and our response to each comment was given in black font.

Comment 1: Figure 2, add more detail to what red and yellow lines represent as well as the red arrow in the Fig. caption.

Response: Thanks very much for your comments. More detailed information about the lines and arrows is added in the figure captions. Fig. 2 displays the SEM images of different BMGs after pre-plastic deformation. In order to make it clear, the figure was updated, and the corresponding caption was also revised to be “Figure 2. Overall morphologies of the (a) Zr56Co28Al16, (d) Zr46Cu46Al8, and (g) Fe48Cr15Mo14Y2C15B6 BMGs after pre-plastic deformation and the enlarged SEM images of the regions (b) R1, (c) R2, (e) R3, (f) R4, (h) R5, and (i) R6, respectively; The solid green lines represent the plastic deformation zone induced by PPD treatments. The yellow dot-dash lines represent the semi-circular shear bands developed from the initial shear bands induced by PPD treatments. In contrast, the red dotted lines represent the induced semi-circular shear bands during the later stage of PPD treatments. The solid red arrows represent the dense wavy shear bands between semi-circular ones, and the dotted yellow arrows represent micro-cracks.”. Moreover, the revised figure was found in the attachment.

Comment 2: XRD analysis was mentioned in the methodology but results not presented. To confirm that, the fabricated BMGs before and after treatment are still amorphous, XRD spectra must be included. This is very critical!.

Response: Thanks very much for your comments. The authors are sorry that we forgot to put the XRD patterns in the supplementary materials. Now combining the suggestions from comment 3, both XRD and EDX observations were added to the new Figure 1(see the attachment). Obviously, all the samples after surface treatments are still amorphous (Fig. 1(b)). In fact, previous reports have demonstrated that nanocrystallization only appears in severely deformed samples or some specific glass-forming alloy systems. For Zr-, ZrCu-, and Fe-based BMGs, the present surface treatments are not fierce enough to induce nanocrystallization.

Comment 3: EDX, XRF or XPS analysis must be included to confirm the elemental compositions of each BMGs.

Response: Thanks for the suggestions. Now the EDX results were added in Figure 1, and it can be seen that the observed BMGs are chemical homogeneous at a micro-scale. The chemical compositions were also measured and shown in the revised manuscript.

Comment 4: Check line 5 “…. due to theesultant …..” kindly check the spelling.

Response: Thanks for the comments. The English was carefully improved. The Abstract and conclusions were also revised.

The authors would like to appreciate the valuable comments from the referees and editors again for improving our manuscript.

Reviewer 2 Report

·         Include the results in abstract section.

·         Kindly include recent literatures in introduction section.

·         Mark the attributes in fig.2

·         Include the error bar in fig. 3

·         Kindly include the standard deviation in table 1.

·         Mark the attributes in fig. 4

·         Conclusion is confusing. Reframe the section as point by point

Recommendation: Major Revision

Author Response

Dear Editors and Reviewers,

The authors sincerely thank the editors again for dealing with our manuscript. We also highly appreciate the reviewers’ constructive comments and valuable suggestions. Those comments are very valuable and helpful for improving our work. In the following, the comments were listed in italic blue font and our response to each comment was given in black font.

Comment 1: Include the results in abstract section.

Response: Thanks very much for your comments. Now the Abstract was revised as follows. Pre-plastic deformation (PPD) treatments on bulk metallic glasses (BMGs) have previously shown to be helpful in producing multiple shear bands. In this work, the applicability of the PPD approach on BMGs with different Poisson’s ratios was validated based on experimental and simulation observations. It was found that for BMGs with high Poisson’s ratios (HBMGs, e.g., Zr56Co28Al16 and Zr46Cu46Al8), the PPD treatment can easily trigger a pair of large plastic deformation zones con-sisting of multiple shear bands. These PPD-treated HBMGs clearly display improved strength and compressive plasticity. On the other hand, the mechanical properties of BMGs with low Poisson’s ratios (LBMG, e.g., Fe48Cr15Mo14Y2C15B6) become worse due to a few shear bands and micro-cracks in extremely small plastic deformation zones. Additionally, for the PPD-treated HBMGs with similar high Poisson’s ratios, the Zr56Co28Al16 BMG exhibits much larger plasticity than the Zr46Cu46Al8 BMG. This phenomenon is mainly due to more defective icosahedral clusters in the Zr56Co28Al16 BMG, which can serve as nucleation sites for shear transformation zones (STZs) during subsequent deformation. The present study may provide a basis for understanding the plastic deformation mechanism of BMGs.

Comment 2: Kindly include recent literatures in introduction section.

Response: Thanks very much for your comments. The recent literature related to bulk metallic glasses and thier composites were updated.

Comment 3: Mark the attributes in fig.2.

Response: Thanks very much for your comments. More detailed information about the lines and arrows is added in the figure captions. Fig. 2 displays the SEM images of different BMGs after pre-plastic deformation. In order to make it clear, the figure was updated, and the corresponding caption was also revised to be “Figure 2. Overall morphologies of the (a) Zr56Co28Al16, (d) Zr46Cu46Al8, and (g) Fe48Cr15Mo14Y2C15B6 BMGs after pre-plastic deformation and the enlarged SEM images of the regions (b) R1, (c) R2, (e) R3, (f) R4, (h) R5, and (i) R6, respectively; The solid green lines represent the plastic deformation zone induced by PPD treatments. The yellow dot-dash lines represent the semi-circular shear bands developed from the initial shear bands induced by PPD treatments. In contrast, the red dotted lines represent the induced semi-circular shear bands during the later stage of PPD treatments. The solid red arrows represent the dense wavy shear bands between semi-circular ones, and the dotted yellow arrows represent micro-cracks.”. Please see the attachment

Comment 4: Include the error bar in fig. 3.

Response: Thanks very much for your comments. The compression test was repeated for three times. The error bars of the data obtained from Fig. 3 (see the attachment) have been added in Table 1. According to the suggestions, the error bars were also added in the figure.

Comment 5: Kindly include the standard deviation in table 1.

Response: Thanks very much for your comments. In fact, the error bar include in the Table 1 is the corresponding deviation.

Comment 6: Mark the attributes in fig. 4.

Response: Thanks very much for your comments. The corresponding attributes were marked. Moreover, the solid arrows and dotted square represent shear bands and the regions where shear bands exist (see the attachment) .

Comment 7: Conclusion is confusing. Reframe the section as point by point.

Response: Thanks very much for your comments. The conclusion was carefully revised, which was given as follows. In this work, three types of BMGs with different Poisson’s ratios, i.e., Zr46Cu46Al8, Zr56Co28Al16, and Fe48Cr15Mo14Y2C15B6 BMGs, were chosen to verify the impact of the pre-plastic deformation on the generation of multiple shear bands in BMGs. According to the XRD and EDS observations, all the samples are still amorphous, and no nanocrystal-lization appears after the pre-plastic deformation via the current sideloading method. For the Zr46Cu46Al8 and Zr56Co28Al16 BMGs with high Poisson’s ratios, a pair of larger plastic deformation zones can be induced at loading positions after pre-plastic deformation. Many long semi-circular shear bands appear within both plastic deformation zones, while some dense wavy shear bands emerge between these long shear bands. The domi-nant reason is that the resultant force changes continuously when the pre-plastic defor-mation proceeds. For the Fe48Cr15Mo14Y2C15B6 BMG with a high Poisson’s ratio, only very small plastic deformation zones are caused after the pre-plastic deformation. Within these small zones, only a few semi-circular shear bands appear, accompanied by several micro-cracks.

Then the room-temperature compression tests were conducted on these pre-deformed BMG rods to measure their mechanical properties. It was observed that those BMGs with high Poisson’s ratios show higher strength and larger plasticity after pre-plastic defor-mation. However, the Fe48Cr15Mo14Y2C15B6 BMG with a high Poisson’s ratio after the pre-plastic deformation exhibits worse mechanical properties than the as-cast state, which should be attributed to the relatively small STZ volume because of a low Poisson’s ratio. Additionally, despite the Poisson’s ratios of the Zr56Co28Al16 and Zr46Cu46Al8 BMGs are similar, the improvement of mechanical properties of the Zr56Co28Al16 BMG is more evi-dent than those of the Zr46Cu46Al8 BMG. Molecular dynamics simulation observations in-dicate that the Co- and Al-centered architectures in the as-cast Zr56Co28Al16 BMG contain more defective icosahedral clusters, which can provide more nucleation sites for STZs during deformation and lead to better mechanical properties.

The authors would like to appreciate the valuable comments from the referees and editors again for improving our manuscript.

Reviewer 3 Report

1. The abstract part should be revised.

2. Figure 7 needs modification, the author should modify this figure in order to make it more readable to readers.

3. Improve the English grammar of the manuscript.

Author Response

Dear Editors and Reviewers,

The authors sincerely thank the editors again for dealing with our manuscript. We also highly appreciate the reviewers’ constructive comments and valuable suggestions. Those comments are very valuable and helpful for improving our work. In the following, the comments were listed in italic blue font and our response to each comment was given in black font.

Comment 1: The abstract part should be revised.

Response: Thanks very much for your comments. Now the Abstract was revised as follows. Pre-plastic deformation (PPD) treatments on bulk metallic glasses (BMGs) have previously shown to be helpful in producing multiple shear bands. In this work, the applicability of the PPD approach on BMGs with different Poisson’s ratios was validated based on experimental and simulation observations. It was found that for BMGs with high Poisson’s ratios (HBMGs, e.g., Zr56Co28Al16 and Zr46Cu46Al8), the PPD treatment can easily trigger a pair of large plastic deformation zones con-sisting of multiple shear bands. These PPD-treated HBMGs clearly display improved strength and compressive plasticity. On the other hand, the mechanical properties of BMGs with low Poisson’s ratios (LBMG, e.g., Fe48Cr15Mo14Y2C15B6) become worse due to a few shear bands and micro-cracks in extremely small plastic deformation zones. Additionally, for the PPD-treated HBMGs with similar high Poisson’s ratios, the Zr56Co28Al16 BMG exhibits much larger plasticity than the Zr46Cu46Al8 BMG. This phenomenon is mainly due to more defective icosahedral clusters in the Zr56Co28Al16 BMG, which can serve as nucleation sites for shear transformation zones (STZs) during subsequent deformation. The present study may provide a basis for understanding the plastic deformation mechanism of BMGs.

Comment 2:  Figure 7 needs modification, the author should modify this figure in order to make it more readable to readers.

Response: Thanks very much for your comments. The data in Figure 7 was obtained from simulation. The people draw them like us or convert the X axis and Y axis of the graph. The authors are not kidding and really do not know how to make it better. Then both kinds of figures are provided here for option (see the attachment). In the revised manuscript, the firs one was used. The authors hope that the reviewer can understand us.

Comment 3:  Improve the English grammar of the manuscript.

Response: Thanks very much for your comments. Now the English was carefully polished, and the corresponding revision was marked in the revised manuscript.

The authors would like to appreciate the valuable comments from the referees and editors again for improving our manuscript.

Round 2

Reviewer 1 Report

Add the EDS spectrum in Figure 1, not only mapping.

Author Response

Since the EDS point spectrum for each BMG was repeated at least ten times, all the EDS point spectrum was shown in Figs. S1-S3 in the supplementary material.

Reviewer 2 Report

I am afraid to read the manuscript. Kindly remove the strict lines and resend for review.

Author Response

The authors were informed that any revisions made to the manuscript should be marked up using the “Track Changes” function if you are using MS Word. We are sorry for the troubles.  The second version should be better now.

Round 3

Reviewer 2 Report

The author made all the corrections. The latest version is adequate for publication.

Author Response

Thanks a lot for the recommendation.